# The Molecular and Functional Characterization of Sensory Neuron Membrane Protein 1b (SNMP1b) from *Cyrtotrachelus buqueti* (Coleoptera: Curculionidae)

**DOI:** 10.3390/insects15020111

**Published:** 2024-02-04

**Authors:** Hua Yang, Long Liu, Fan Wang, Wei Yang, Qiong Huang, Nanxi Wang, Hongling Hu

**Affiliations:** National Forestry and Grassland Administration Key Laboratory of Forest Resources Conservation and Ecological Safety on the Upper Reaches of the Yangtze River, College of Forestry, Sichuan Agricultural University, Chengdu 611130, China; liulong@sicau.edu.cn (L.L.); 2019204010@stu.sicau.edu.cn (F.W.); 10625@sicau.edu.cn (W.Y.); 12062@sicau.edu.cn (Q.H.); wangn4182@sicau.edu.cn (N.W.); 13979@sicau.edu.cn (H.H.)

**Keywords:** *Cyrtotrachelus buqueti*, sensory neuron membrane protein, expression pattern, binding assay, molecular docking

## Abstract

**Simple Summary:**

The bamboo weevil beetle, *Cyrtotrachelus buqueti* (Coleoptera: Curculionidae) is the primary bamboo pest and seriously influences the development of the bamboo industry. Sensory neuron membrane proteins (SNMPs) play important roles in insect pheromone communication. However, SNMPs of *C. buqueti* are still uncharacterized. With the aim of developing control techniques to control this pest, one novel *SNMP* from *C. buqueti*, *CbuqSNMP1b*, was functionally characterized in this study. The results indicated that *CbuqSNMP1b* was predominantly expressed in the antennae of both sexes and showed significantly higher transcription levels in the adult stage, suggesting that *CbuqSNMP1b* is involved in the process of olfaction. Fluorescence binding assays revealed that CbuqSNMP1b could bind to three out of fourteen volatile compounds emitted from *C. buqueti* and showed the strongest binding affinity to dibutyl phthalate, demonstrating the role of SNMP1 in detecting odorants. Molecular docking was performed to further understand the binding mode between CbuqSNMP1b and these three ligands, and the results showed that hydrophobic interactions were the prevailing forces within the binding cavities of CbuqSNMP1b. The results of this study will be helpful to understand the function of SNMP1 in *C. buqueti* and lay a foundation for developing new methods to control this pest.

**Abstract:**

Sensory neuron membrane proteins (SNMPs) play important roles in insect chemoreception and SNMP1s have been reported to be essential in detecting sex pheromones in *Drosophila* and some lepidopteran species. However, SNMPs for *Cyrtotrachelus buqueti* (Coleoptera: Curculionidae), a major insect pest of bamboo plantations, remain uncharacterized. In this study, a novel *SNMP* gene, *CbuqSNMP1b*, from *C. buqueti* was functionally characterized. The expression of *CbuqSNMP1b* was significantly higher in antennae than in other tissues of both sexes and the expression level was significantly male-biased. Additionally, *CbuqSNMP1b* showed significantly higher transcription levels in the adult stage and very low transcription levels in other stages, suggesting that *CbuqSNMP1b* is involved in the process of olfaction. Fluorescence binding assays indicated that CbuqSNMP1b displayed the strongest binding affinity to dibutyl phthalate (*K*_i_ = 9.03 μM) followed by benzothiazole (*K*_i_ = 11.59 μM) and phenol (*K*_i_ = 20.95 μM) among fourteen *C. buqueti* volatiles. Furthermore, molecular docking revealed key residues in CbuqSNMP1b that interact with dibutyl phthalate, benzothiazole, and phenol. In conclusion, these findings will lay a foundation to further understand the olfactory mechanisms of *C. buqueti* and promote the development of novel methods for controlling this pest.

## 1. Introduction

Insects perceive their environment based primarily on the semiochemicals, which provide cues to seek mates and host plants, and avoid predators. Therefore, chemoreception is of crucial importance for the survival and reproduction of most insect species [1,2]. Insects have evolved sophisticated olfactory sensilla to perceive olfactory cues and the sensilla are generally located on the antennae [2,3]. In insect olfactory perception, when hydrophobic odorants enter the sensillum lymph via cuticular pores, odorant binding proteins (OBPs) or chemosensory proteins (CSPs) in the sensilla lymph can bind to the odorants and deliver them to receptors (e.g., odorant receptors, ORs, and ionotropic receptors, IRs) located in the dendritic membrane of olfactory sensory neurons (OSNs), thus guiding the insect behavior [2,4]. In addition, it was demonstrated that another protein family in dendritic membranes of OSNs, sensory neuron membrane proteins (SNMPs), play critical roles in odorant perception [5].

Insect SNMPs are homologs of the vertebrate CD36 protein superfamily, which serve as transporters of cholesterol, lipids, and fatty acids and receptors of lipoproteins [6,7,8,9]. CD36 proteins possess two transmembrane domains and a large extracellular domain, which plays a crucial role in ligand interaction [10,11]. The first SNMP was discovered in the olfactory neurons of the polyphemus moth, *Antheraea polyphemus*, and named ApolSNMP1 [6]. Subsequently, variable numbers of SNMPs across insect species and orders have been identified [8,12,13,14]. Based on phylogenetic relationships, insect SNMPs have been divided into four groups (SNMP1–SNMP4) [13,15]. 

Since SNMP1 and SNMP2 subtypes were found in all insect species studied so far, studies on the expression and function of SNMPs in the olfactory system have concentrated on these two subtypes [12]. Studies on *Drosophila melanogaster* demonstrated that DmelSNMP1 was required to detect the sex pheromone 11-cis-vaccenyl acetate (cVA) [16,17]. Furthermore, it was found that the rapid activation and termination of cVA-induced activity depended on SNMP1 [18]. In addition, SNMP1 was shown to interact with the sex pheromone receptor and played critical roles in sex pheromone detection in lepidopteran insects [19,20]. The SNMP2 subtype was first characterized from *Manduca sexta* and shares 27% identity with SNMP1 [21]. In some lepidopteran insects, SNMP1s were mainly expressed in adult antennae [6,21,22], while SNMP2s were wildly expressed in multiple tissues [13]. Less is known about the function of SNMP2s compared to SNMP1s. However, the distinct expression profiles suggest that SNMP1 and SNMP2 are likely to play different roles in chemoreception.

The bamboo weevil beetle, *Cyrtotrachelus buqueti* (Coleoptera: Curculionidae) is the primary bamboo pest and mainly distributed in bamboo-producing areas around the Sichuan Basin in Guangdong, Guangxi, Chongqing, Guizhou, and Shanghai provinces in China as well as in Thailand, Myanmar, Vietnam, and other Southeast Asian countries [23]. These beetles mainly damage several bamboo species, including *Bambusa textilis*, *Phyllostachys pubescens*, *Neosinocalamus affinis*, and *Dendrocalamus farinosus* [24]. This pest damages bamboo shoots via piercing and sucking, as well as laying eggs, thereby hindering the development of bamboo products and causing significant economic losses [25]. For this reason, this insect pest has been listed as a dangerous forestry pest by the State Forestry Administration of China since 2003 [26]. Olfactory proteins including SNMPs are desirable molecular targets for developing novel pest control methods. However, little is known about the SNMPs from this species and other coleopteran insects.

In the present study, one *SNMP* gene, *CbuqSNMP1b*, from *C. buqueti* was functionally characterized. The expression patterns of *CbuqSNMP1b* in different tissues of both sexes and at different developmental stages were evaluated. Using the bacterial expression system, the CbuqSNMP1b protein was successfully expressed and purified. And then, with the purified recombinant protein, the binding properties of CbuqSNMP1b to fourteen volatiles emitted by *C. buqueti* were tested with fluorescence competitive binding assays. Finally, the molecular modeling and molecular docking were carried out to facilitate the understanding of the molecular interaction mechanisms between CbuqSNMP1b and ligands. This research will contribute to the investigation of the olfactory mechanism of *C. buqueti* and establish a theoretical foundation for the development of novel pest control methods.

## 2. Materials and Methods

### 2.1. Sequence and Phylogenetic Analysis

*CbuqSNMP1b* was retrieved from a previously annotated transcriptome (GenBank accession number: SRS1876730) [27]. The NCBI ORF Finder (https://www.ncbi.nlm.nih.gov/orffinder, accessed on 2 December 2023) was used to predict the open reading frame (ORF) of *CbuqSNMP1b*. Putative transmembrane domains of CbuqSNMP1b protein were predicted with the TMHMM program (https://services.healthtech.dtu.dk/services/TMHMM-2.0/, accessed on 3 December 2023) and finally modeled with Protter 1.0 (https://wlab.ethz.ch/protter/start/, accessed on 2 December 2023). The molecular weight (MW) and isoelectric point (pI) of CbuqSNMP1b were determined using the Compute pI/MW program on the Expasy server (https://web.expasy.org/protparam/, accessed on 2 December 2023). Multiple sequence alignments were performed using the MAFFT method [28] with the auto algorithm and BLOSUM62 scoring matrix and were edited using the ESPript 3.0 (http://espript.ibcp.fr/ESPript/ESPript/, accessed on 3 December 2023).

For the phylogenetic analysis, amino acid sequences of CbuqSNMP1b and other coleopteran SNMPs were aligned using the MAFFT method as described above. Based on the Bayesian information criterion (BIC), the best-fit model of amino acid evolution for coleopteran SNMPs was determined with the ModelFinder integrated in PhyloSuite (version 1.2.2) [29,30]. The best-fit model for coleopteran SNMPs was LG + F + R3. Under the best-fit model, the maximum likelihood (ML) tree of coleopteran SNMPs was constructed using IQ-TREE [31] in PhyloSuite (version 1.2.2) with the default parameters. Bootstrap support of tree branches was assessed with 5000 ultrafast bootstraps [32], as well as the Shimodaira–Hasegawa-like approximate likelihood-ratio test (SH-aLRT) with 1000 replicates [33]. The phylogenetic tree was visualized by using the iTOL server (https://itol.embl.de/, accessed on 3 December 2023).

### 2.2. Insect Rearing and Tissue Collection

Pupae of *C. buqueti* were collected from the bamboo planting base (30°13′ E, 102°91′ N) in Lushan County, Ya’an City, Sichuan Province, China, and maintained in our laboratory at 25 ± 1 °C, 70% ± 10% relative humidity, and a 14:10 (L:D) photoperiod. The newly emerged adults were reared on fresh *N. affinis* shoots in insect cages (60 × 60 × 60 cm) under the same conditions. For tissue-specific gene expression, male and female individuals were segregated and dissected to antennae, heads (without antennae), thoraxes, abdomens, wings, and legs. One sample contained at least 5 individual tissues and more than five sample replicates were prepared. For development stage-specific gene expression, whole bodies of female adults, male adults, pupae, old larvae (5-instars), young larvae (2-instars), and eggs were collected, respectively. All samples were immediately frozen in liquid nitrogen and then preserved at −80 °C prior to the analysis.

### 2.3. Total RNA Extraction, cDNA Synthesis, and Quantitative Real-Time PCR (qPCR)

Following the manufacturer’s instructions, total RNA extraction was performed by using a MiniBEST Universal RNA Extraction Kit (TaKaRa, Beijing, China). The RNA integrity was verified with gel electrophoresis. The quality (OD_260/280_ > 1.8) and concentration of total RNA samples were determined with the DU800 spectrophotometer (Beckman Coulter, Brea, CA, USA). According to the manufacturer’s protocol, 1 µg of the total RNA was used to synthesize first-strand cDNA using a Goldenstar™ RT6 cDNA Synthesis Kit Ver.2 (Beijing TsingKe Biotech Co., Ltd., Beijing, China). All cDNA samples were preserved at −20 °C until further use.

The qPCR analysis was performed to determine the tissue-specific and development-stage-specific expression of *CbuqSNMP1b*. The cDNA was diluted 1:5 in sterilized PCR-grade water as a template for the qPCR analysis. The glyceraldehyde 3-phosphate dehydrogenase (*GAPDH*) gene (GenBank accession number: KY745870.1) was used as the endogenous reference to normalize the target gene expression. The primers (Appendix A) of target and reference genes were designed with Primer Premier Software (version 5.0). The melting curve analysis was used to validate the specificity of each primer. The efficiency of each primer pair was calculated by analyzing the standard curve with a 10-fold cDNA dilution series. Using the TB Green™ Premix Ex Taq™ (Tli RNaseH Plus) Kit (TaKaRa, Beijing, China) as DNA-binding fluorescence dye, the qPCR was performed on the CFX96™ Real-time PCR detection system (Bio-Rad, Hercules, CA, USA). Each qPCR reaction was conducted in triplicates and the volume of each reaction was 25 µL, containing 8.5 µL of sterilized PCR-grade water, 12.5 µL of TB Green Premix Ex Taq II (Tli RNaseH Plus), 1 µL each of sense and antisense primers (10 µM), and 2 µL of sample cDNA. All test samples were analyzed in three biological replicates. The qPCR cycling parameters were as follows: an initial denaturation for 30s at 95 °C, followed by 40 cycles of 95 °C for 5 s and 60 °C for 30 s. The 2^−ΔΔCt^ method [34] was used to determine relative levels of gene expression.

### 2.4. Preparation of Recombinant CbuqSNMP1b Protein

The large extracellular loop ectodomains of CbuqSNMP1b were amplified from plasmids containing the full-length coding regions using the specific primers (Appendix A). According to the instructions of a ClonExpress II One Step Cloning Kit (Vazyme, Nanjing, China), the PCR products were cloned into pET-28a (+) bacterial expression vectors through homologous recombination. The cloned fragment was verified via sequencing. The recombinant plasmid was transformed into Rosetta (DE3) competent cells for recombinant protein expression. The verified bacterial suspension was inoculated into a liquid LB medium containing 50 μg/mL of kanamycin and incubated at 37 °C until cells reached an optical density at 600 nm (OD_600_) of 0.6–0.8. Then, protein expression was induced with the addition of isopropyl β-D-thiogalactoside (IPTG) to a final concentration of 0.1 mM. Cells were further incubated for 8 h at 30 °C and the bacteria were harvested through centrifugation at 16,000 rpm for 50 min at 4 °C. The pelleted bacterial cells were resuspended in a cooled NTA-0 buffer containing lysozymes (final concentration of 0.1 mg/mL) and then exposed to ultrasonication. The soluble fraction and the insoluble pellet were analyzed with 12% sodium dodecyl sulfate–polyacrylamide gel electrophoresis (SDS-PAGE) using the Unstained Protein Molecular Weight Marker (Solarbio Science & Technology Co., Ltd., Beijing, China) as the marker. The recombinant protein was present in the pellet as inclusion bodies and was precipitated with the STET buffer while DTT (a final concentration of 1 mM) was added with ice ultrasound. Precipitation was repeated until a transparent supernatant was visible. The resulting inclusion bodies were resuspended with a 6 M guanidine hydrochloride solution containing a final concentration of 5 mM DTT. The mixture was incubated at 37 °C with shaking at 220 rpm for 4 h until inclusion bodies had been completely dissolved. The mixture was then centrifuged to obtain the supernatant containing the denatured protein. The protein solution was placed in a dialysis bag and dialyzed with the NTA-0 buffer for 48 h (4 °C). The dialyzed protein was purified using an Ni-NTA His-Bind column (Invitrogen, Waltham, MA, USA) with imidazole washing at gradient concentrations of 20, 60, 200, and 500 mM. This step is followed by dialysis to remove imidazole and then the final protein was dissolved in 50 mM Tris-HCl. The purified protein was then detected with an SDS-PAGE analysis. The Bradford Protein Assay Kit (Solarbio Science & Technology Co., Ltd., Beijing, China) was used to determine the concentration of the purified protein, and the concentration of the recombinant CbuqSNMP1b protein was 0.5 mg/mL.

### 2.5. Fluorescence Competition Binding Assay

Binding affinities of CbuqSNMP1b to 14 volatiles (Table 1) emitted by female adults of *C. buqueti* were measured with the fluorescence competition binding assays using *N*-phenyl-1-naphthylamine (1-NPN) as the fluorescent probe. 1-NPN and all of the other chemicals (the purity being at least more than 96%) used in the binding assays were purchased from Aladdin Biochemical Technology Co., Ltd. (Shanghai, China). The Lumina fluorescence spectrophotometer (Thermo Fisher Scientific, Waltham, MA, USA) was used to perform fluorescence binding assays with a 1 cm light path quartz cuvette at room temperature. All the slit widths for both excitation and emission were 10 nm. The 1-NPN and all the other chemicals were dissolved in methanol (HPLC-grade) to a final concentration of 1 mM. The binding constant of 1-NPN was determined by titrating a 2 μM solution of the CbuqSNMP1b protein with aliquots of 2–40 μM 1-NPN concentrations. The binding of 1-NPN was measured at an excitation wavelength of 337 nm. The scanning range was 360–560 nm at each concentration and the maximum fluorescence intensities were plotted against the free ligand concentration. Under the assumption that the protein was 100% active and that the stoichiometry of binding was 1:1 (protein/ligand) at saturation, the dissociation constant (*K*_d_) of CbuqSNMP1b with 1-NPN was calculated by performing a Scatchard analysis of the binding data using GraphPad Prism 8.4.1.

Using both CbuqSNMP1b and 1-NPN at a concentration of 2 μM, the binding affinities of CbuqSNMP1b to various volatile ligands were tested through competition assays by adding each competitor ligand at different concentrations. The *K*_i_ value (*K*_d_ of the competitor) was determined based on the IC_50_ value (half maximal inhibitory concentration) using the equation *K*_i_ = [IC_50_]/(1 + [1-NPN]/*K*_1-NPN_), where [1-NPN] is the free concentration of 1-NPN and *K*_1-NPN_ is the *K*_d_ of the protein/1-NPN complex.

### 2.6. Structural Modeling and Molecular Docking

The SWISS-MODEL homology modeling server (http://swissmodel.expasy.org/, accessed on 8 December 2023) was used to construct the three-dimensional (3D) structure of the extracellular loop ectodomain of CbuqSNMP1b. The Ramachandran plot was used to evaluate the stereochemical quality of the model using the pymod3 plugin implemented in PyMOL 2.4.2 (The PyMOL Molecular Graphics System, Schrödinger, LLC, New York, NY, USA). Compounds that were bound to CbuqSNMP1b were selected for docking studies. Three-dimensional structures of these ligands in SDF format were downloaded from the PubChem database (https://pubchem.ncbi.nlm.nih.gov/, accessed on 8 December 2023). The CB-Dock2 online platform (https://cadd.labshare.cn/cb-dock2/, accessed on 8 December 2023) [35] was utilized to perform the molecular docking analysis, and the optimal docking model was selected based on the most negative binding energy. LigPlot^+^ v.2.2.8 was employed to analyze the interactions between the ligand and receptor and draw 2D depictions. PyMOL 2.4.2 was used for model visualization. 

### 2.7. Statistical Analysis

The GraphPad Prism 8.4.1 software was used to perform statistical analyses of qPCR data. Homogeneity of variance was assessed for all data prior to a further analysis. One-way ANOVA followed by a Tukey’s multiple comparison test were used to determine the statistical significance of differences in developmental stage- and tissue-specific expression levels of the *CbuqSNMP1b* gene. Different letters represent a significant difference (*p* < 0.05) while identical letters are not indicative of a significant difference (*p* > 0.05).

## 3. Results

### 3.1. Sequence Analysis of CbuqSNMP1b

The full-length ORF of *CbuqSNMP1b* is 1566 bp and encodes 521 amino acid residues (Figure 1A). The gene sequence of *CbuqSNMP1b* was verified through PCR amplification and Sanger sequencing. The gene sequence of *CbuqSNMP1b* was deposited in GenBank with the accession number OM908752. The calculated isoelectric point of CbuqSNMP1b is 7.51 and the molecular weight is 59.39 kDa. Same as other SNMPs, CbuqSNMP1b possesses two transmembrane domains in proximity to the N- and C-terminal and a large extracellular loop (Figure 1A,B).

### 3.2. Sequence Alignments and Phylogenetic Analysis

As shown in Figure 2, multiple amino acid sequence alignments indicated that CbuqSNMP1b was highly similar to the homologous proteins from other coleopteran species. CbuqSNMP1b shared the highest sequence identity with *Sitophilus zeamais* SNMP1b (62.0%), followed by *Pachyrhinus yasumatsui* SNMP1b (58.0%), *Dendroctonus adjunctus* SNMP1b (54.3%), *Phyllotreta striolata* SNMP1 (47.6%), and *Coccinella septempunctata* SNMP1 (42.9%). Moreover, the sequence alignment unraveled eight conserved cysteine residues, which are located in the ectodomain and may play a critical role for the protein structures and functions.

To explore the relationships among coleopteran SNMPs, a phylogenetic analysis was performed based on CbuqSNMP1b and other SNMP orthologs from Coleoptera (Figure 3). The results showed two distinct subgroups for SNMP1 and SNMP2 and CbuqSNMP1b fell within the SNMP1 subgroup. Consistent with the results of sequence alignments, CbuqSNMP1b shared the closest phylogenetic relationship with *S. zeamais* SNMP1b.

### 3.3. Expression Pattern Analysis of CbuqSNMP1b

The qPCR was performed to investigate the gene expression levels of *CbuqSNMP1b* in various tissues of both male and female adults, including antennae, legs, wings, abdomens, thoraxes, and heads (without antennae). As shown in Figure 4A, *CbuqSNMP1b* was expressed significantly higher in antennae than other tissues in both male and female adults, with very low expression in other tissues. Furthermore, *CbuqSNMP1b* transcripts in the male antennae were approximately 1.27-fold higher than in the female antennae with a significant difference.

The expression levels of *CbuqSNMP1b* at different developmental stages were also examined, including male and female adults, pupae, old larvae (sixth instar), young larvae (second instar), and eggs (Figure 4B). The results indicated that the expression of *CbuqSNMP1b* was developmentally regulated, and showed significantly higher transcription levels in the adult stage and very low transcription levels in egg, larval, and pupal stages. Similar to the tissue expression pattern, the transcript levels of *CbuqSNMP1b* in male adults were significantly higher than in female adults.

### 3.4. Expression and Purification of CbuqSNMP1b Protein

The theoretical molecular weight is about 51.5 kDa for the CbuqSNMP1b fusion protein. SDS-PAGE displayed a target protein band at 45.0–66.2 kDa in inclusion bodies, consistent with the predicted molecular weight of the fusion protein (Figure 5A). After the fusion protein was purified, a single clear band appeared at 45.0–66.2 kDa (Figure 5B). The results indicated that the recombinant CbuqSNMP1b protein had been successfully expressed in Rosetta competent cells.

### 3.5. Fluorescence Competitive Binding Assays

To investigate the function of CbuqSNMP1b protein, binding abilities of CbuqSNMP1b to 14 volatiles emitted by *C. buqueti* were measured via fluorescence competitive binding assays. The dissociation constant (*K*_d_) of the CbuqSNMP1b/1-NPN complex was first determined, which was then used to calculate the dissociation constants (*K*_i_) of CbuqSNMP1b with other volatile compounds. As shown in Figure 6A, CbuqSNMP1b exhibited high binding affinity to 1-NPN and the dissociation constant of the CbuqSNMP1b/1-NPN complex calculated from the Scatchard plot was 13.30 μM. 

Using 1-NPN as a fluorescence probe, the binding abilities of CbuqSNMP1b to 14 volatiles from *C. buqueti* were determined. The competitive binding curves of CbuqSNMP1b with 14 competitor volatiles are shown in Figure 6B–D and the binding affinities (*K*_i_) are shown in Table 1. CbuqSNMP1b could bind to three out of fourteen volatile compounds of *C. buqueti* (dibutyl phthalate, benzothiazole, and phenol). In particular, CbuqSNMP1b exhibited high binding affinity to dibutyl phthalate and benzothiazole with *K*_i_ values of 9.03 and 11.59 μM, respectively, while the binding affinity of CbuqSNMP1b to phenol was relatively weak with a *K_i_* value of 20.95 μM (Table 1).

### 3.6. Structure Modeling and Molecular Docking

The 3D model of the CbuqSNMP1b ectodomain was generated using homology modeling (Figure 7A). The Ramachandran plot analysis showed that 93.2% of the residues were present in the most favored regions, 6.8% were present in the additional allowed regions, and no residue was present in disallowed regions, indicating the accuracy of the predicted CbuqSNMP1b structure (Figure 7B). The eight conserved cysteine residues form four pairs of disulfide bonds (Figure 7A). In the middle of the CbuqSNMP1b protein model, a cavity was formed and was surrounded by α helix, β folding, and random curling, forming a relatively stable structure (Figure 7A).

A molecular docking analysis was carried out to further elucidate the binding mode of CbuqSNMP1b to dibutyl phthalate, benzothiazole, and phenol. The binding energy was used to evaluate the strength of the binding ability of CbuqSNMP1b to these three compounds, with lower binding energy indicating a higher binding capacity. Consistent with *K_i_* values determined from the fluorescence binding assays (Table 1), CbuqSNMP1b showed the best binding features with dibutyl phthalate with a binding energy value of −6.8 kcal/mol, followed by benzothiazole and phenol with binding energy values of −5.1 and −4.4 kcal/mol, respectively. CbuqSNMP1b bound to dibutyl phthalate via 11 hydrophobic contacts and 1 hydrogen bond (Figure 8A). CbuqSNMP1b bound to benzothiazole via five hydrophobic contacts and one hydrogen bond (Figure 8B). CbuqSNMP1b bound to phenol only via nine hydrophobic contacts (Figure 8C).

## 4. Discussion

In the present study, one *SNMP* gene from *C. buqueti*, *CbuqSNMP1b*, was analyzed. The topo structure of CbuqSNMP1b showed typical features of the SNMP family including two transmembrane domains and a large extracellular loop, which is consistent with that of the vertebrate CD36 family [36]. Multiple sequence alignment indicated that CbuqSNMP1b shared high sequence similarities with SNMP1s of other coleopteran insects, suggesting conserved roles of SNMP1 in these species. The large extracellular loop of CbuqSNMP1b contained eight conserved cysteine residues, which may form four stabilizing disulfide bridges and be important for the structure and function of SNMPs [21]. The phylogenetic analysis demonstrated that CbuqSNMP1b belonged to the SNMP1 subfamily.

The expression pattern analysis revealed that *CbuqSNMP1b* was mainly expressed in adult antennae of both sexes with a higher expression level in male antennae than in females, which is similar to previous reports on lepidopteran, hymenopteran, and dipteran insects [37,38,39], suggesting that CbuqSNMP1b may be specifically involved in olfaction and may play critical roles in male chemoreception. The tissue and sex expression pattern of *CbuqSNMP1b* was also consistent with that of two pheromone-binding proteins (PBPs) of *C. buqueti* [40,41]. Additionally, the expression level of *CbuqSNMP1b* was highly developmentally regulated and *CbuqSNMP1b* showed high transcription levels in the adult stage, which was similar to that of *CsupSNMP1* from *Chilo suppressalis* (Lepidoptera: Pyralidae) [42], indicating that *CbuqSNMP1b* plays important roles in chemosensory behavior of adults. The expression levels of olfactory proteins, such as SNMPs, are likely to synchronize with the production of female sex pheromones, enhancing male sensitivity to behaviorally active sex pheromones and thereby influencing olfactory-modulated behaviors in moths [22]. The SNMP1 is involved in the detection of sex pheromones in *D. melanogaster* [16,17] and lepidopteran insects [13,19,20]. Therefore, it was speculated that CbuqSNMP1b might be related to recognizing sex pheromones based on the above results.

Data from *Drosophila* showed that SNMP1 as a coreceptor might be involved in the transfer of pheromones or other odorants from OBPs to their cognate ORs via the binding of ligands to its large, tunnel-like ectodomain [11]. To investigate the olfactory function of CbuqSNMP1b, the fluorescence competition binding assay was performed to determine the binding affinities of CbuqSNMP1b to 14 volatiles released from female adults of *C. buqueti*. The results showed that CbuqSNMP1b could bind to dibutyl phthalate, benzothiazole, and phenol, indicating the role of *CbuqSNMP1b* in detecting odorants. Consistent with the binding abilities of CbuqPBP1 and CbuqPBP2 to dibutyl phthalate, which may be the major component of the *C. buqueti* sex pheromone [40,41], CbuqSNMP1b displayed the strongest binding ability with dibutyl phthalate, suggesting that CbuqSNMP1b may play an important role in the process of detecting and transferring the sex pheromone. Although CbuqSNMP1b could also bind to benzothiazole and phenol, there is no behavioral evidence to demonstrate the exact function of these two chemicals in the olfactory perception of *C. buqueti*. Further study is still required to explore the role of the interaction between CbuqSNMP1b and these two chemicals in the olfactory perception of *C. buqueti*. It is well known that insect SNMPs play important roles in the transmembrane transport of lipophilic compounds [12]. In the present study, CbuqSNMP1b could bind aromatics, suggesting that insect SNMP1s may have a broad ligand spectrum.

The functionality of SNMP1 is believed to rely on the formation of disulfide bonds facilitated by conserved cysteines in its extracellular domain [12]. In previous studies, the extracellular domains of most insect SNPM1s were predicted to form three pairs of disulfide bonds from six conserved cysteine residues, such as *D. melanogaster* SNMP1 and *A. polyphemus* SNMP1 [11,21]. In a recent study, it was found that four disulfide bonds were formed in the extracellular domain of *Helicoverpa armigera* SNMP1 [20]. Similar to *H. armigera* SNMP1, the eight conserved cysteine residues in the extracellular domain of CbuqSNMP1b form four pairs of disulfide bonds, which may make the structure of SNMP1 more stable. By performing a molecular docking analysis, the binding mode of CbuqSNMP1b to dibutyl phthalate, benzothiazole, and phenol was determined to further understand the molecular interaction mechanisms. The binding strength in molecular docking was evaluated using the change in potential energy surrounding the binding site during the interaction of the protein and ligand [43]. In line with fluorescence binding assays, docking results showed the lowest binding energy with dibutyl phthalate, followed by benzothiazole and phenol, and hydrophobic interactions were the prevailing forces within the binding cavities of CbuqSNMP1b. In addition, hydrogen bonds were also involved in the interaction of CbuqSNMP1b with dibutyl phthalate and benzothiazole, which was the reason why CbuqSNMP1b showed high binding abilities to these two compounds. Additionally, compared to dibutyl phthalate and benzothiazole, the amino acid residues of CbuqSNMP1b interacting with phenol were obviously different, further indicating that insect SNMP1 may have a broad ligand spectrum.

## 5. Conclusions

In conclusion, CbuqSNMP1b seems to perform essential functions in chemoreception, particularly in the detection of sex pheromones in *C. buqueti*. The information provided in this study will be helpful for further functional study of SNMP1 in this insect and lay a theoretical foundation for the development of new pest control strategy by interfering with olfactory perception.

## Figures and Tables

**Figure 1 insects-15-00111-f001:**
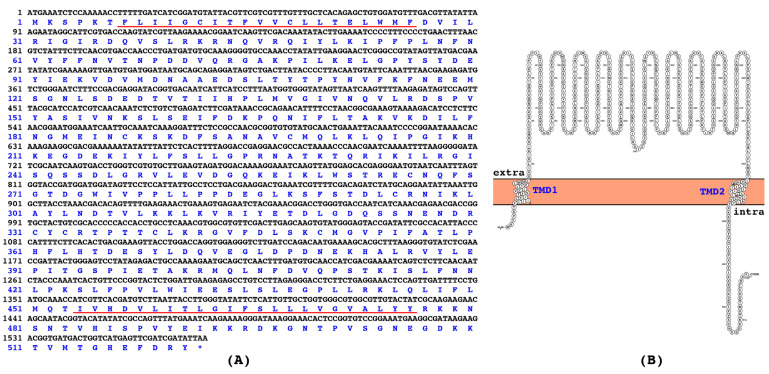
(**A**) Nucleotide and deduced amino acid sequences of *CbuqSNMP1b*. The stop codon is indicated by an asterisk, and the two putative transmembrane domains are underlined. (**B**) The transmembrane domain structure of CbuqSNMP1b. TMD indicates the transmembrane domain.

**Figure 2 insects-15-00111-f002:**
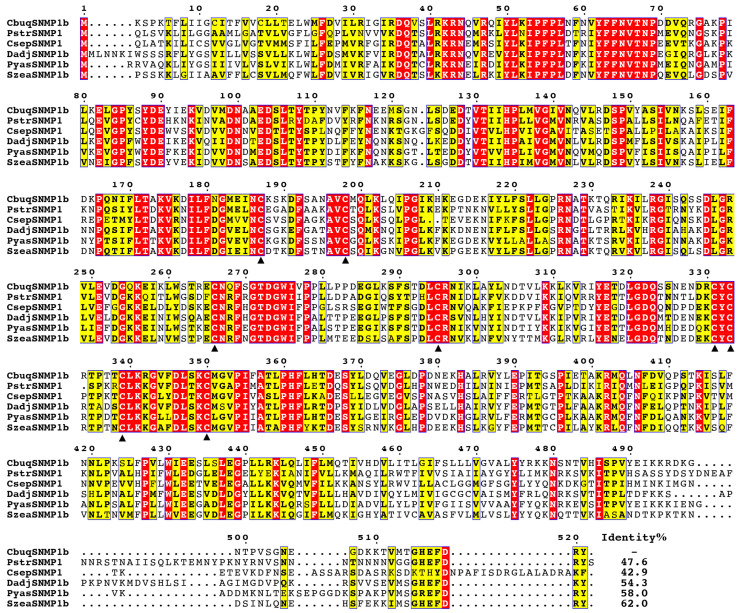
Multiple amino acid sequence alignments of CbuqSNMP1b with SNMP1 from other coleopteran insects. The eight conserved cysteine residues are highlighted by black triangles. The abbreviated species names and GenBank accession numbers of amino acid sequences are as follows: *Cyrtotrachelus buqueti* (CbuqSNMP1b, WAQ79968.1); *Phyllotreta striolata* (PstrSNMP1, ANQ46504.1); *Coccinella septempunctata* (CsepSNMP1, XP_044749433.1); *Dendroctonus adjunctus* (DadjSNMP1b, QKV34982.1); *Pachyrhinus yasumatsui* (PyasSNMP1b, WJJ63367.1); *Sitophilus zeamais* (SzeaSNMP1b, QEX08000.1).

**Figure 3 insects-15-00111-f003:**
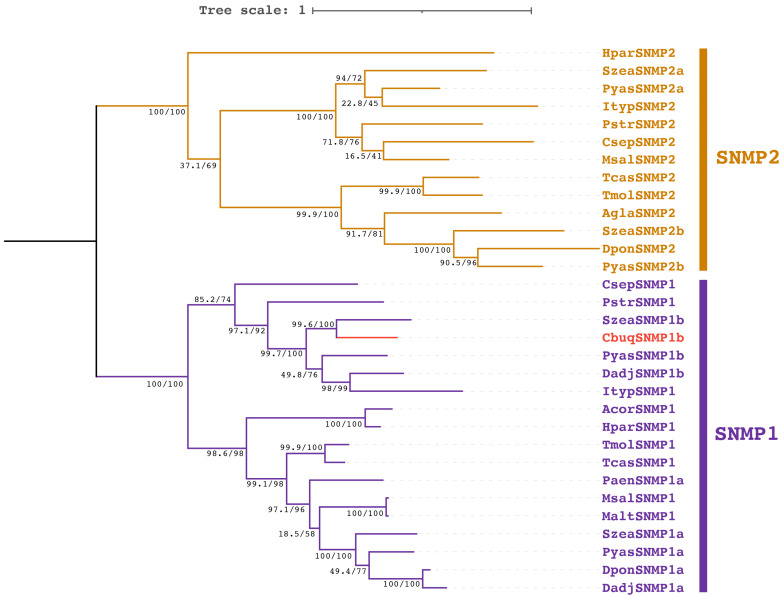
Phylogenetic analysis of coleopteran SNMPs using the maximum likelihood method. Bootstrap support values and SH-aLRT values are indicated on branches. The tree is rooted via a midpoint approach. CbuqSNMP1b is highlighted in red. The species and GenBank accession numbers of all selected SNMPs are as follows: *Cyrtotrachelus buqueti* (CbuqSNMP1b, WAQ79968.1); *Holotrichia parallela* (HparSNMP1, AVM18969.1; HparSNMP2, AVM18970.1); *Sitophilus zeamais* (SzeaSNMP1a, QEX07999.1; SzeaSNMP1b, QEX08000.1; SzeaSNMP2a, QEX08001.1; SzeaSNMP2b, QEX08002.1); *Pachyrhinus yasumatsui* (PyasSNMP1a, WJJ63366.1; PyasSNMP1b, WJJ63367.1; PyasSNMP2a, WJJ63368.1; PyasSNMP2b, WJJ63369.1); *Ips typographus* (ItypSNMP1, JAA74404.1; ItypSNMP2, JAA74403.1); *Phyllotreta striolata* (PstrSNMP1, ANQ46504.1; PstrSNMP2, ANQ46505.1); *Coccinella septempunctata* (CsepSNMP1, XP_044749433.1; CsepSNMP2, XP_044745604.1); *Monochamus saltuarius* (MsalSNMP1, QUP79609.1; MsalSNMP2, QUP79610.1); *Tribolium castaneum* (TcasSNMP1, XP_001816436.2; TcasSNMP2, XP_970008.1); *Tenebrio molitor* (TmolSNMP1, AJO62245.1; TmolSNMP2, AJO62246.1); *Anoplophora glabripennis* (AglaSNMP2, XP_018566911.1); *Dendroctonus ponderosae* (DponSNMP1a, AGI05171.1; DponSNMP2, XP_019770844.2); *Dendroctonus adjunctus* (DadjSNMP1a, QKV34981.1; DadjSNMP1b, QKV34982.1); *Anomala corpulenta* (AcorSNMP1, AKC58519.1); *Pyrrhalta aenescens* (PaenSNMP1a, APC94303.1).

**Figure 4 insects-15-00111-f004:**
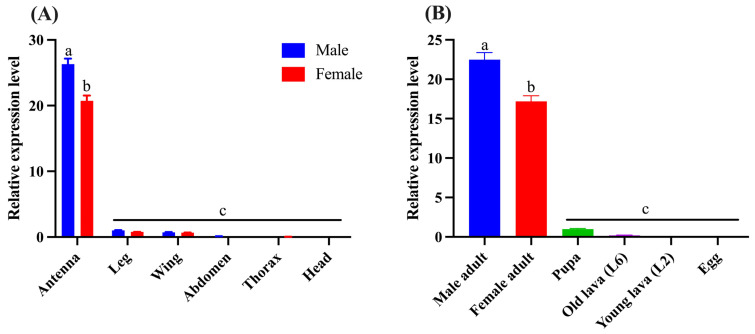
Expression patterns of *CbuqSNMP1b* in various tissues and developmental stages. (**A**) Tissue expression profiles of *CbuqSNMP1b*. (**B**) Expression levels of *CbuqSNMP1b* at different developmental stages. The L2 and L6 represent the second- and sixth-instar larvae, respectively. All data are presented as the means ± standard deviation (SD) with three biological replicates. Different lowercase letters above the bars denote significant differences (*p* < 0.05) among different tissues or stages.

**Figure 5 insects-15-00111-f005:**
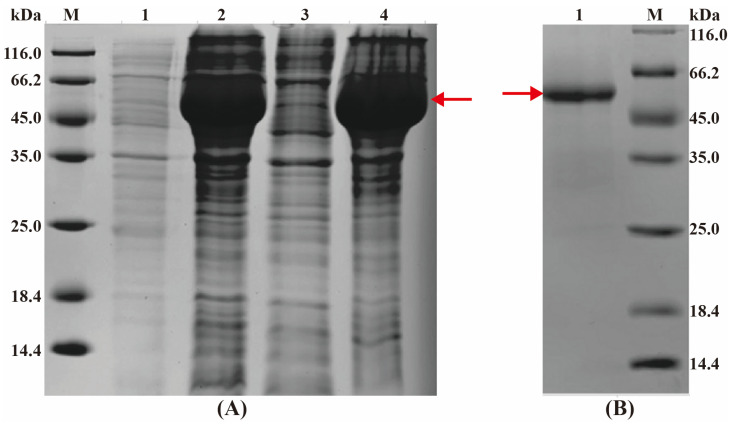
Expression and purification of CbuqSNMP1b protein. (**A**) SDS-PAGE analysis of the recombinant CbuqSNMP1b protein. M: protein marker; Lane 1: non-induced bacterial culture lysed with loading buffer; Lane 2: IPTG-induced bacterial culture lysed with loading buffer; Lane 3: the soluble supernatant after ultrasound; Lane 4: inclusion bodies after ultrasound. (**B**) SDS-PAGE analysis of purified recombinant CbuqSNMP1b. Lane 1: purified fusion protein after renaturation. The target protein is indicated by the red arrow.

**Figure 6 insects-15-00111-f006:**
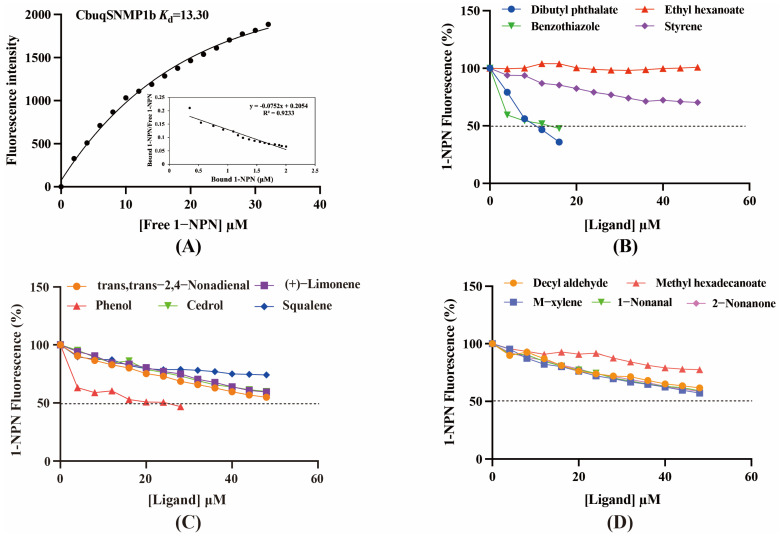
Competitive fluorescence binding assays of CbuqSNMP1b. (**A**) The binding curve and Scatchard plot of 1-NPN with CbuqSNMP1b. (**B**–**D**) Competitive binding curves of CbuqSNMP1b to fourteen volatiles emitted by *C. buqueti*. The dotted line indicates that half of 1-NPN is replaced by the ligand.

**Figure 7 insects-15-00111-f007:**
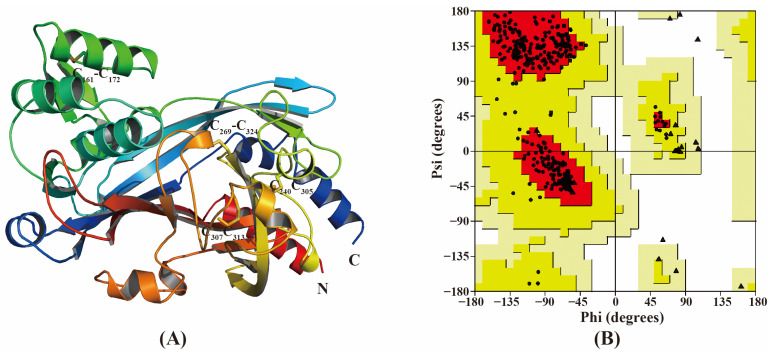
(**A**) Three-dimensional (3D) structural model of CbuqSNMP1b. Disulfide bonds are indicated by C_161_-C_172_, C_269_-C_324_, C_240_-C_305_, and C_307_-C_313_. (**B**) Ramachandran plot analysis. The most favorable regions are colored by red. Additional allowed, generously allowed and disallowed regions are indicated as yellow, light yellow and white fields, respectively. Circles represent non-glycine and non-proline residues. Triangles and squares represent glycine and proline residues, respectively.

**Figure 8 insects-15-00111-f008:**
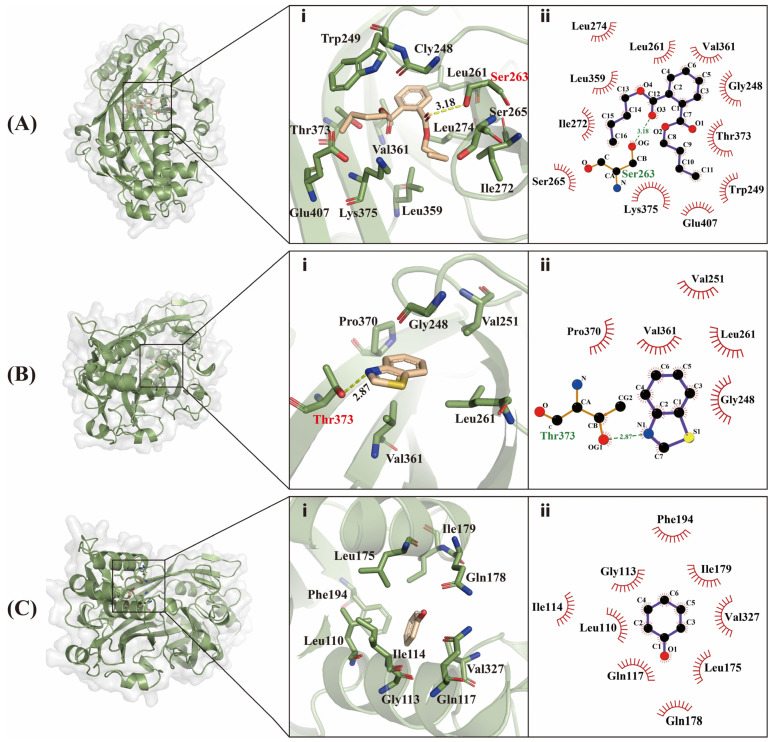
Competitive binding modes of CbuqSNMP1b with dibutyl phthalate (**A**), benzothiazole (**B**), and phenol (**C**). (**i**) Three-dimensional demonstrations of the binding interface. (**ii**) Two-dimensional demonstrations of the detailed binding of the key residues with volatile compounds.

**Table 1 insects-15-00111-t001:** Binding affinities of CbuqSNMP1b with fourteen volatiles emitted by *C. buqueti*.

Ligand	Structural Formula	CAS No.	IC_50_ (μM)	*K*_i_ (μM)
Dibutyl phthalate	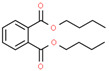	84-74-2	10.39	9.03
Ethyl hexanoate	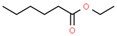	123-66-0	― ^1^	―
Benzothiazole	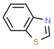	95-16-9	13.33	11.59
Styrene	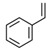	100-42-5	―	―
trans,trans-2,4-Nonadienal	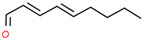	5910-87-2	―	―
(+)-Limonene	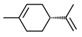	5989-27-5	―	―
Phenol	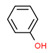	108-95-2	24.10	20.95
Cedrol	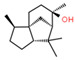	77-53-2	―	―
Squalene	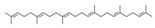	111-02-4	―	―
Decyl aldehyde	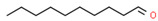	112-31-2	―	―
M-xylene	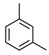	108-38-3	―	―
Methyl hexadecanoate	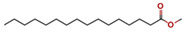	112-39-0	―	―
1-Nonanal	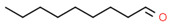	124-19-6	―	―
2-Nonanone	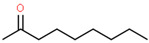	821-55-6	―	―

^1^ It was considered that the protein had no binding with the tested ligands if the IC_50_ values > 50 μM and *K*_i_ values were not calculated, and these two values are represented as “―”.

## Data Availability

Data are contained within the article and Appendix A.

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
