# Peer review of "The Molecular and Functional Characterization of Sensory Neuron Membrane Protein 1b (SNMP1b) from Cyrtotrachelus buqueti (Coleoptera: Curculionidae)"

_insects, 2024, doi:10.3390/insects15020111_

Round 1

Reviewer 1 Report

Comments and Suggestions for Authors

In their manuscript “Molecular and Functional Characterization of Sensory Neuron Membrane Protein 1b (SNMP1b) from Cyrtotrachelus buqueti (Coleoptera: Curculionidae)”, the authors identified and functionally characterized the SNMP1b type of the bamboo weevil beetle. Through qPCR experiments, the expression level of CbuqSNMP1b was found to be highest in adult male antennae followed by adult female antennae. Very low transcription levels were found in other appendages as well as younger developmental stages. By using the heterologously expressed and purified CbuqSNMP1b ectodomain in a 1-NPN-based competitive binding assay, the authors found binding of the protein to three C. buqueti body odors out of 14 tested. In addition, the authors evaluated the 3D structure of the protein and assessed by molecular docking approaches the interaction of the three odorants in the protein’s ectodomains. Taken together, the authors propose that CbuqSNMP1 appears to play an essential function in chemoreception, particularly in the detection of putative pheromone components in the beetle C. buqueti.

The authors present interesting and new data regarding the role of SNMP1b in chemoreception that are of particular interest to people in olfaction but also interesting for a wider audience. Generally, the manuscript is well written and structured. The data are generally well presented. However, there are a number of major and minor points that need to be addressed prior to acceptance of the manuscript for publication.

major points and suggestions:

1) Some of the references used in the introduction are not appropriate:

- in lines 49-51,  the authors talk about how sensilla perceive olfactory cues, but the reference used [3] is a study on expression levels of SNMPs on adult citrus flies; in lines 51-56, which talks about the IRs, ORs, OBPs and CSPs, the study that was referenced [4] is a study that focuses on SNMPs, not on the mentioned proteins; in lines 64-65, the authors refer to the various numbers of SNMPs across insect species, yet the reference used [12] is a study about SNMPs in only the desert locust. Perhaps include more sources or add the review you already cited [15].

2) As this is the first time that an SNMP1 ectodomain has been purified and used in a functional assay, more detailed information must be given in the “Material and Methods” section. Please address the following points in section 2.4 “Preparation of recombinant CbuqSNMP1b”:

Line 179: specify - in what buffer was the bacterial pellet suspended for ultrasonification?

How was the so-called “inclusion bodies” fraction prepared? This is not specified. Additionally, proteins in inclusion bodies are usually denatured. What did you do to reconstitute or check your purified protein’s tertiary structure?

Line 182: what brand was the Nickel NTA column? And what were the concentrations of the imidazole gradient? Final concentration?

How was the protein purified from the imidazole? In which buffer was the purified protein finally solved?

And: in section 2.5 “Fluorescensce Competion Binding assay”:

What was the assay buffer used in the competitive 1-NPN binding assay?

3) The authors suggest that CbuqSNMP1b forms 4 disulfide bridges resulting from 8 cysteine residues, which is more than what has been found in other homologs. In D. melanogaster SNMP1 (Gomez-Diaz et al 2016), in moth SNMP1 (Rogers et al 2001) and in mammalian CD36 proteins (Glatz and Luikken 2018), the proteins only form 3 disulfide bridges. Please address this difference in the discussion.

4) In the discussion a role of CbuqSNMP1 in sex pheromone detection is suggested, due to its higher expression in males. In this context, the authors should indicate from which sex the body odors you tested are released. Also, it is stated that dibutyl phthalate may be a putative pheromone component. What is known about the biological function of the other two substances for which binding to SNMP1b was found? Please indicate.

5) The identified ligands of CbuqSNMP1 exhibit chemical properties quite different from SNMP1 ligands indicated for moths and the fruit fly. The odorants found in this study are aromatics, whereas in moths and flies, the SNMP1 ligand are long-chain aliphatics. This suggest that SNMP1s from different species may have different ligand spectra or that SNMP1s generally might have a very broad ligand spectrum. It would improve the discussion to include this aspect into the discussion.

In this context, the modelling data indicate binding of the three identified SNMP1b ligands to partially different sites of the ectodomains (Figure 8, panel C shows different amino acids compared to panels B and A). It would also be worth to briefly discuss this finding.

6) Figure 1: the second transmembrane domain region shown in A does not match B. In Figure 1B the the amino acids I, V, H are included in the TMD region, which are not underlined in Figure1A. This should be corrected so that the image is coherent. Also, to improve the clarity in Figure 1B: consider adding “TMD” in front the “1” and “2”, to indicate the transmembrane regions and better place this label right, respectively and left to the TMDs. In the moment it reads like “Extra1”.

6) In Figure 6: the scaling (y-axis) for “D” is different compared to “B” and “C”. For consistency of data presentation, please use the same scaling for D as in B and C. Also include the dotted 50 % line in D and indicate it meaning in the Legend of Figure 6.

Minor points

The manuscript contains a number of grammar and spelling mistakes as well as few other minor points that need to be addressed

-          Line 29: change “pheromone” to “pheromones”

-          Line 57: “play critical roles role” Remove the second role.

-          Line 74: “share” should be changed to “shares”

-          Line 94: Change “bacteria” to “bacterial”

-          Line 97: Change “was” to “were”.

-          Lines 107-108: Change “predicted” to “predict”

-          Line 112: your abbreviation for molecular weight should be consistent. Change “Mw” to “MW” to match it on line 113.

-          Line 137; Line 150: Consider changing “stages-specific” to “stage-specific”

-          Line 138, please specify which organs you used from female adults, male adults, pupae, old larvae, young larvae. Or specific if you used the whole body.

-          Line144: Change “OD260/280” to “OD260/280” to match line 176 (OD600)

-          Line 161: change “triplicate” to “triplicates”

-          Line 181: change “SDSPAGE” to “SDS-PAGE”

-          Line 195: you wrote “dissolved in in methanol”. Remove one “in”

-          Line 201: add “the” before “protein”

-          Line 226: change “were” to “was”

-          Line 228: change “tests” to “test”

-          Line 234: change “encode” to “encodes”

-          Line 239: change “possess” to “possesses”

-          Line 257: Change “sub group” to “subgroup”

-          Line 285: Change “investigated” to “investigate”

-          Line 311: Your write “Above results” when referring to your SDS-PAGE, but due to the formatting of the manuscript, the results come after this part in the text. Consider rephrasing to just “The results”.

-          Figure 5, figure legend: please specify more clearly the fractions that you applied onto the lanes.

-          Line 384: remove “the” from “also consistent the with that”

-          Lines 389-392: this sentence is difficult to understand. Please rephrase.

-          Line 392: change “pheromone” to “pheromones”

-          Line 395: add “be” after “might”

-          Line 395: change “involve” to “involved”

Reviewer 2 Report

Comments and Suggestions for Authors

              The reviewer has read with much interest the manuscript entitled as ‘Molecular and Functional Characterization of Sensory Neuron Membrane Protein 1b (SNMP1b) from Cyrtotrachelus buqueti (Coleoptera: Curculionidae)’ submitted by Hua Yang, Long Liu, Fan Wang, Wei Yang, Qiong Huang, Nanxi Wang and Hongling Hu to INSECTS. The authors comprehensively investigated SNMP1b of Cyrtotrachelus buqueti on the molecular level and found that it performs essential functions in olfaction of this insect. The manuscript is fundamentally well written but some problems are present in the manuscript as follows.

Lines 48-49       Therefore, chemoreception is of crucial importance for insect survival and reproduction --> (opinion) As the authors stated, chemoreception is of crucial importance for survival and reproduction of insect, but since most dragonflies have poor olfactory function, this account is not applicable for such species of insects.

Line 53              chemosensory protein (CSPs) --> (question) What is differences between the chemosensory protein and the binding protein?

Line 55              thus guide the insect --> (question) Are odorant binding proteins (OBPs) or chemosensory proteins (CSPs) nominative in this sentence? They are probably inadequate.

Line 122            LG + F + R3--> what is this?

Line 155            primer premier software --> Primer Premier Software

Line 182            NTA His-Bind column --> Please add the maker etc.

Line 267            Figure 3 is not referred and not explained in the text.

Line 285            investigated --> investigate

Lines 327-328   Using 1-NPN as a fluorescence probe, the binding abilities of CbuqSNMP1b to 14 volatiles from C. buqueti were determined --> (question) Why did the authors examine only the volatiles from C. buqueti and did not examine other general odors?

Line 347            (Figure 5B) --> Is it exact?  Figure 7b may be true.

Lines 379-384   The expression patterns analysis revealed that CbuqSNMP1b was mainly expressed in adult antennae of both sexes with higher expression level in male antennae than in females, which was similar to previous reports in lepidopteran, hymenopteran and dipteran insects [4,37,38], suggesting that CbuqSNMP1b may be specifically involved in olfaction and play critical roles in male chemoreception. The tissue and sex expression pattern of CbuqSNMP1b was also consistent the with that of two pheromone binding proteins (PBPs) of C. buqueti. --> (opinion) CbuqSNMP1b was expressed higher in male than female. The reviewer thinks possibly that this fact may stem largely from larger number of olfactory sensilla in male than in female and that the CbuqSNMP1b may be related to most olfactory sensilla and not be restricted to the pheromone receptive sensilla. This inference is consistent with the fact that CbuqSNMP1b is abundant not only in male antennae but also in female antennae and is few in other parts (organs).

Lines 417-418   In conclusion, CbuqSNMP1b seems to perform essential functions in chemoreception, particularly in the detection of sex pheromones in C. buqueti. --> (opinion) As stated above, the function of CbuqSNMP1b is not restricted to pheromone reception processes and CbuqSNMP1b may work in all or most olfactory reception processes. 
